# How to catch a shear band and explain plasticity of metallic glasses with continuum mechanics

Oleksandr Glushko [1] ✉, Reinhard Pippan[2], Daniel Şopu [2], Christian Mitterer[1] & Jürgen Eckert[1,2]

Capturing a shear band in a metallic glass during its propagation experimentally is very challenging. Shear bands are very narrow but extend rapidly over macroscopic distances, therefore, characterization of large areas at high magnification and high speed is required. Here we show how to control the shear bands in a pre-structured thin film metallic glass in order to directly measure local strains during initiation, propagation, or arrest events. Based on the experimental observations, a model describing the shear banding phenomenon purely within the frameworks of continuum mechanics is formulated. We claim that metallic glasses exhibit an elastic limit of about 5% which must be exceeded locally either at a stress concentrator to initiate a shear banding event, or at the tip of a shear band during its propagation. The model can successfully connect micro- and macroscopic plasticity of metallic glasses and suggests an alternative interpretation of controversial experimental observations.

In contrast to crystalline metals, where structural defects, such as dislocations, grain boundaries, or stacking faults, act as carriers of plasticity, the major plasticity mechanism in amorphous metallic alloys is rapid and highly localized shear displacement, leading to the appearance of shear bands (SBs). Despite intensive research, understanding the fundamental mechanisms responsible for the initiation and operation of SBs remains a long-standing challenge due to partially contradictory experimental observations that were reported in the literature and are briefly discussed below.

One subject of debate is the temperature rise inside SBs. Direct experimental evidence of temperature increase within SBs[1–4] supported an assumption that during a shear banding event, the temperature might exceed the glass transition temperature ($T_g$), bringing the material into a viscous state, which enables high shear displacements without rupture. However, it was convincingly demonstrated by several research groups that SBs can also stay cold, suggesting that heating is rather a consequence of rapid plastic deformation than the cause of it[5–11]. Another point of discussion is the SB propagation mode. Shear displacement was shown to occur simultaneously along the whole SB plane[8,9,12,13] but, at the same time, progressive propagation of SBs was reported in many cases[1,2,14–16]. The width of SBs is typically measured to be within 10–20 nm[17–19], however, strong indications of material properties alterations as far as tens of micrometers away from a shear band were reported[20–22]. Also, the structure of the SBs appears to be many-faced: post-mortem high-resolution transmission electron microscopy (HRTEM) studies reported on barely detectable structural changes within a SB[23,24], both increase and decrease of material density along a SB[25], local reduction of density[26], chemical segregation[23], or even formation of cavities within SBs[27,28].

Although the various possibilities of SB operation, as well as the potential variety of structural material states within SBs, are at least well-documented, the process of SB initiation remains a major ongoing issue. According to the pioneer work of Argon[29] there exist elementary events of atomic-scale shear displacements called shear transformation zone (STZ). Based on this concept, it is hypothesized that percolation, or collective activation, of many STZs might be the condition for initiation of a SB[30]. Although a direct experimental proof of this

[1]Department of Materials Science, Montanuniversität Leoben, Leoben, Austria. [2]Erich Schmid Institute of Materials Science, Austrian Academy of Sciences, Leoben, Austria. ✉e-mail: oleksandr.glushko@unileoben.ac.at

hypothesis has not been delivered thus far, computer simulations are widely employed for studying STZ activation and SB formation[30–32].

Simultaneously with the wide acceptance of the STZ concept, a hypothesis that a certain degree of local material softening through local free volume gain (also called structural rejuvenation or dilatation) should occur to enable SB generation and propagation is frequently mentioned in the literature. In earlier models, this dilatation is supposed to stem from adiabatic heating within the SBs[1,3,33]. During the last decade, the models assuming cold rejuvenation (alternatively called mechanical dilatation or shear dilatation) were suggested based on the observation that mechanical stress can cause similar structural changes as the high temperature[9,26,34–37] in MGs. In contrast to the collective activation of STZs, the cold dilatation model assumes that autocatalytic structural excitation first creates a band of softened material while the actual shear displacement propagates within this band afterward. It is worth noting that direct experimental evidence proving that cold rejuvenation indeed occurs prior to SB formation has never been provided. The task of resolving local material structure in nm-scale during a shear banding event is hampered by the difficulty of obtaining precise atomic- and nano-scale structural information for an amorphous material. Therefore, because of the fast dynamics and localized nature of shear banding, the mechanism of plastic deformation in MGs cannot be properly resolved by experimental means yet.

To sum up, the current situation with understanding fundamental mechanisms behind the initiation and operation of SBs in metallic glasses reminds on the old parable about blind men and an elephant: a large number of carefully measured properties are reported (involving highly sophisticated and well-though experiments), but the whole phenomenon is still a puzzle. One of the missing pieces of the puzzle is the information about the local strain state within and in the vicinity of an SB during its formation and propagation. Several attempts of using digital image correlation (DIC) to acquire the distribution of mechanical strain on the surface of a strained sample did not bring sufficiently new insights due to lacking spatial resolution of DIC analysis[16,38,39] or missing intermediate stages of SB propagation[40]. A recent in-situ nanobeam electron diffraction study reported on local strain measurements within a nano-sized notched MG specimen[41]. Despite outstanding spatial resolution of strain maps and provided detailed analysis of local structural changes, only sudden sample fracture without indication of distinct shear banding events was described.

In this work, we demonstrate how to "freeze" the dynamic stages of SB propagation and map full-field strains with the spatial resolution of a few tens of nanometers. Numerous SB initiation and SB arrest events are recorded and analyzed, enabling the quantitative formulation of local conditions for SB operation and the establishment of a continuum mechanics model of shear banding.

## Results

### Experimental approach

When the tensile strain, applied to a polymer-supported thin film MG (see Fig. 1a and Methods section for more details), reaches the global elastic limit of approximately 2%, the cracks start to form and propagate quickly as described in a recent paper[42]. These cracks are formed through activation of out-of-plane SBs, i.e., when shear displacement occurs within a plane perpendicular to the surface[42]. In the current study, we are aiming at capturing the propagation of in-plane SBs, i.e., those with the shear displacement within the film plane. Therefore, the films were pre-structured using Focused Ion Beam (FIB) to prevent crack formation within the region of interest (ROI). The three FIB patterns employed in this work for observation of propagating SBs are called stochastic (S), notch (N), and notch-arrest (N-A), according to the sketch shown in Fig. 1b. Two straight and parallel to each other line cuts oriented perpendicularly to the

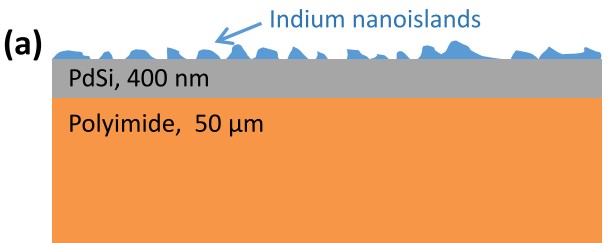

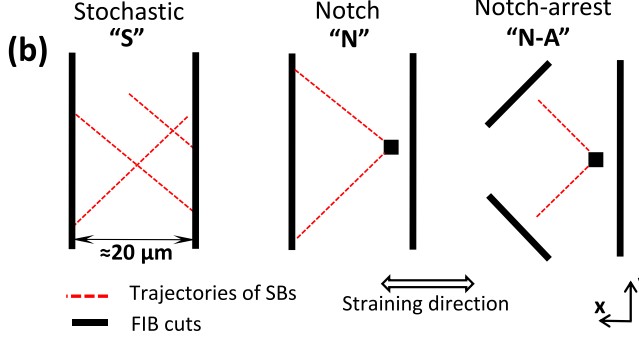

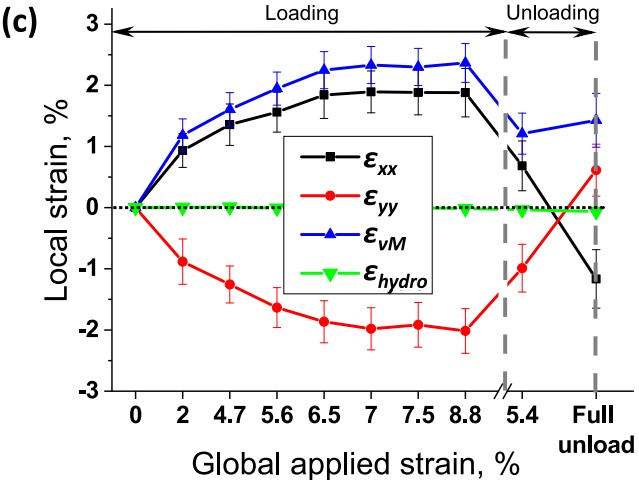

**Fig. 1 | Description of the experimental approach. (a)**: the schematics of the considered materials system; (**b**): the schematics of the three FIB patterns used to pre-structure the film; (**c**): dependence of the local principal strain components ($\varepsilon_{xx}$ and $\varepsilon_{yy}$) as well as local von Mises strain ($\varepsilon_{vM}$) and hydrostatic strain ($\varepsilon_{hydro}$) on the applied global strain between two vertical FIB cuts (pattern S). The error bars depict the standard deviations.

straining direction should lead to reduced major strain component ($\varepsilon_{xx}$), but not significantly affecting the minor strain component ($\varepsilon_{yy}$) within the ROI. This should lead to the prevention of crack propagation but allow stochastic in-plane SB formation (pattern S). With an additional notch in the form of a square-shaped FIB cut, the SBs are expected to be generated from the corners of the cut due to stress concentration (pattern N). The third pattern is designed in a way that generated from the notch SBs are forced to propagate towards a zone where both major and minor strain components tend to zero thus providing a condition for SB arrest (pattern N-A).

Due to the pre-patterning, the local strain tensor within ROIs deviates significantly from the global applied (uniaxial) strain. In this paper, we consider a mechanical loading profile consisting of seven global loading steps (0% – 2% – 4.7% – 5.6% – 6.5% – 7% – 7.5% – 8.8% of total global strain) and two unloading steps (5.7 % – fully unloaded) applied in-situ in a scanning electron microscope (SEM). The measured relationship between the globally applied strain and local strains

within the area between the two cuts of pattern S is given in Fig. 1c. The film is essentially in a 2D strain state where the principal strains ($\varepsilon_{xx}$ and $\varepsilon_{yy}$) have similar magnitudes and opposite signs so that the hydrostatic in-plane strain $\varepsilon_{hydro} = (\varepsilon_{xx} + \varepsilon_{yy})/2$ is close to zero during deformation. Another specific feature of the considered system is that in the fully unloaded state, the principal strains change their signs and are non-zero. This is caused by viscoelastic relaxation of the polymer substrate leading to compressive major strain and tensile minor strain after full unloading. The correspondence between global and local strains shown in Fig. 1c should be kept in mind when the shear banding events are discussed in detail below.

### Local strain around a propagating shear band

Dynamic states of SB propagation are shown in Fig. 2a–d on the example of pattern type N (see Fig. 1b). The shear band was initiated at the corner of the square notch and, at the loading step two (4.7% of total global strain), its tip just entered the ROI at the point $A$, as shown in Fig. 2a. The distribution of von Mises strain along the line connecting points $A$ and $B$ is depicted in Fig. 2e (black curve). At this stage, the strain at point $A$ is about 5% decreasing gradually to the average value of about 1.5%. Within the next loading step, the SB extended over the ROI towards point $B$ still having a tip characterized by gradually decreasing strain (Fig. 2b). Within the next loading steps, the shear band developed further, reaching a mature state at the highest applied global strain of 8.8% (Fig. 2c). Von Mises strain within the SB is measured to be between 25 and 35%, still having a gradient between the points $A$ and $B$ according to Fig. 2e, blue curve. At the final state after full unloading (Fig. 2d), the von Mises strain within the ROI is non-zero due to the viscoelastic relaxation of the substrate which induces compressive major strain and tensile minor strain, as was demonstrated in Fig. 1c. Within the SB, homogeneous strain relaxation after unloading is captured in Fig. 2e (compare the blue and purple curves). It is important to note, that the slip occurs within a narrow plane with a thickness of 10−20 nm which is of the order of pixel size (13 nm) and lower than the subset size and step used for DIC (195 nm and 65 nm, respectively, see Methods section for more details). Therefore, the apparent width of the high-strain bands in Fig. 2a–d does not directly

correspond to the width of a shear band but includes all subsets that were influenced by the slip.

The average von Mises strains for eight different SBs from the considered patterns (see Fig. 1b) subjected to the same loading profile are shown in Fig. 2f. Different SBs have different histories reflecting the stochastic nature of shear banding events. The SBs from the S patterns (i.e., in the absence of artificial stress concentrators) were generated at the last loading step and, therefore, exhibited lower values of von Mises strain. In patterns with artificial stress concentrators (N and N-A), the SBs were initiated at lower applied global strains and have experienced multiple shear banding events leading to higher values of total von Mises strain. At the same time, all SBs demonstrate similar elastic relaxation behavior upon unloading.

Having captured the dynamic states of SB evolution, one can estimate how much shear displacement is associated with a single shear banding event. In Fig. 2g, h, SEM images of the area corresponding to the dotted square in Fig. 2d are shown for the initial and the fully unloaded state, respectively. The surface trace marked by dashed red lines enables the measurement of the absolute value of total displacement which is about 140 nm. Five loading steps (not counting the step shown in Fig. 2a) caused shear displacement within the considered SB giving an average of 28 nm per loading step. Since multiple shear banding events might have occurred during a single loading step, the shear displacement associated with a single shear banding event might be significantly lower in the range of a few nanometers. Although this estimation cannot be directly generalized over different MGs and, more importantly, over different sample geometries, experimental setups, and loading profiles, it is clear that the detection of a shear displacement as small as 28 nm is below the resolution capabilities of many reported in-situ experiments[1,12,14,16,38,43,44]. Thus, the development of premature SBs might have remained unnoticed. The video showing the evolution of the three components of the strain tensor ($\varepsilon_{xx}$, $\varepsilon_{yy}$, and $\varepsilon_{xy}$) with increasing global strain for the ROI considered in Fig. 2 is provided online as a Supplementary Video 1. The corresponding raw SEM images are available as Supplementary Data.

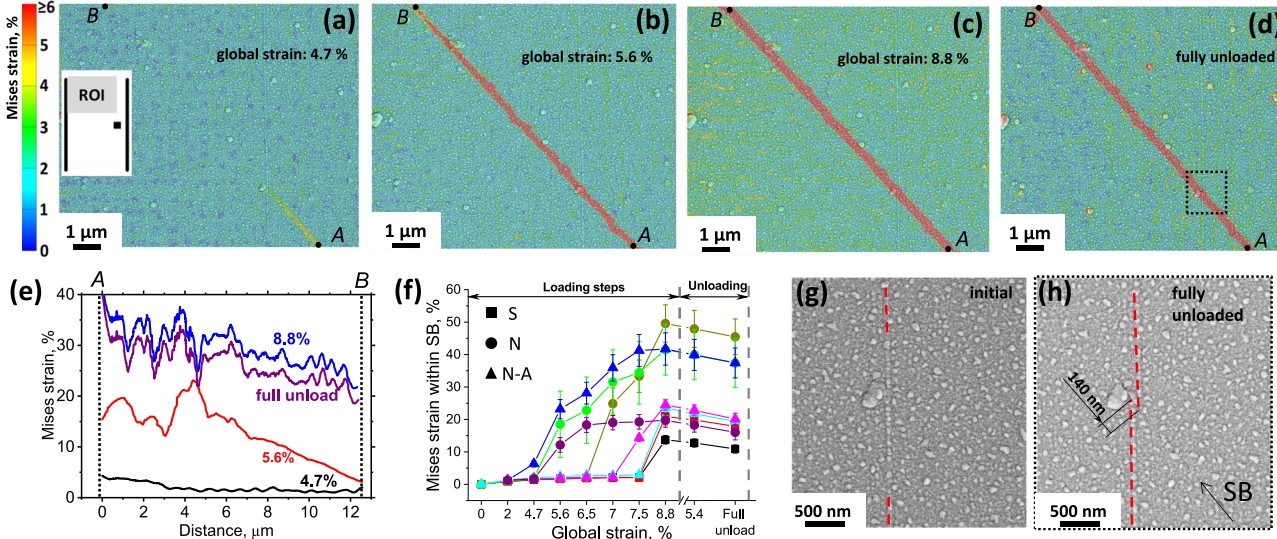

**Fig. 2 | Capturing the dynamics of SB propagation.** The von Mises strain maps of different stages of SB propagation are shown in (**a–d**) with the corresponding global strain state defined on the figures. The position of ROI is sketched in the inset of (**a**). The distributions of von Mises strain along a line connecting points A and B are shown in (**e**), with each curve designated by a corresponding global strain state. The evolution of the average von Mises strain within eight different SBs depending on the applied global strain for loading and unloading is given in (**f**). The different symbols correspond to the three FIB patterns according to Fig. 1b. The error bars depict the standard deviations. SEM images comparing the area corresponding to the dotted square in (**d**) before loading and after full unloading are shown in (**g**) and (**h**), respectively. The position of the SB is marked by the arrow in (**h**). The red dashed lines in (**g, h**) mark a pre-existing surface trace to highlight the shear displacement caused by the shear banding events. The straining direction is horizontal.

## Birth and arrest of shear bands

The N-A patterns (see Fig. 1b) are shown to be effective in arresting the propagating SBs: the SBs are not able to penetrate through the strain relaxation zones, generated by the inclined FIB cuts. Out of four considered N-A patterns, in three cases the SB was initiated, as expected, from the corner of the square. In one case, however, two SBs were generated away from the milled pattern enabling strain mapping of a valuable event of unnotched SB birth. The von Mises strain maps after the initiation of the first SB, the second SB as well as after full unloading are shown in Fig. 3a–c, respectively. Figure 3a–c are based on SEM images with a pixel size of 26 nm, while the areas inside the white dashed rectangles were analyzed using magnified images with a pixel size of 13 nm. The video showing all three independent components of the strain tensor at each loading step is available as Supplementary video 2. At applied global strain of 7% (Fig. 3a) the first SB was generated (marked by I in Fig. 3a), however, its propagation was hindered leading to decaying strain on its both ends. At the last loading step with an applied global strain of 8.8%, another shear banding event occurred within this SB, leading to its spatial extension, and a new SB (marked by II in Fig. 3b) was generated. The strain map of the same area after unloading is shown in Fig. 3c. Since DIC analysis captures total surface displacements, unambiguous separation of the measured strain into elastic and plastic parts is not possible during loading. The fact that SB II was arrested in the middle of ROI during the last loading step offers a convenient opportunity to quantify reversible and irreversible deformation within an SB tip by comparing the local strains at maximum applied load and after unloading. The magnified areas around the tip

of the SB II are shown in Fig. 3d–f. The three black circles marked by $A$, $B$, and $C$ indicate three characteristic points: within the SB, within the SB tip, and ahead of the SB tip, respectively. By comparing von Mises strain distributions at maximum strain (Fig. 3e) and after unloading (Fig. 3f), it might appear that within the tip of the SB (i.e., between the points $A$ and $B$) the strain was of elastic nature, i.e., it recovered after unloading. However, the von Mises strain does not account for the change in the sign of the principal strains which occurs after unloading (as discussed in Fig. 1c). The line profiles of $\varepsilon_{xx}$, $\varepsilon_{yy}$, and von Mises strain between points $A$, $B$, and $C$ are depicted in the corresponding Fig. 3g–i for the same global loading steps as shown in Fig. 3d–f, respectively. As one can see, after unloading the $\varepsilon_{xx}$ and $\varepsilon_{yy}$ strain components change their sign at point "B" and have virtually constant values between points $B$ and $C$ (Fig. 3i). Such a state corresponds to the global unloaded state of the film indicating that no plastic deformation occurred. However, between points A and B, the major strain component $\varepsilon_{xx}$ remained positive, and the minor strain component $\varepsilon_{yy}$ negative, indicating permanent plastic deformation. Such a detailed analysis is performed in order to underline the difference between SBs and cracks with respect to the strain localization around the tip. The whole fracture mechanics is based on the fact that the stress concentration at the crack tip governs the crack propagation dynamics, depending on the applied load and material properties. In contrast to crack formation, shear banding is a plastic phenomenon, i.e., after the arrest of a propagating SB the material remains intact, therefore, no permanent strain concentration associated with the tip is detected within the resolution capabilities of the applied method.

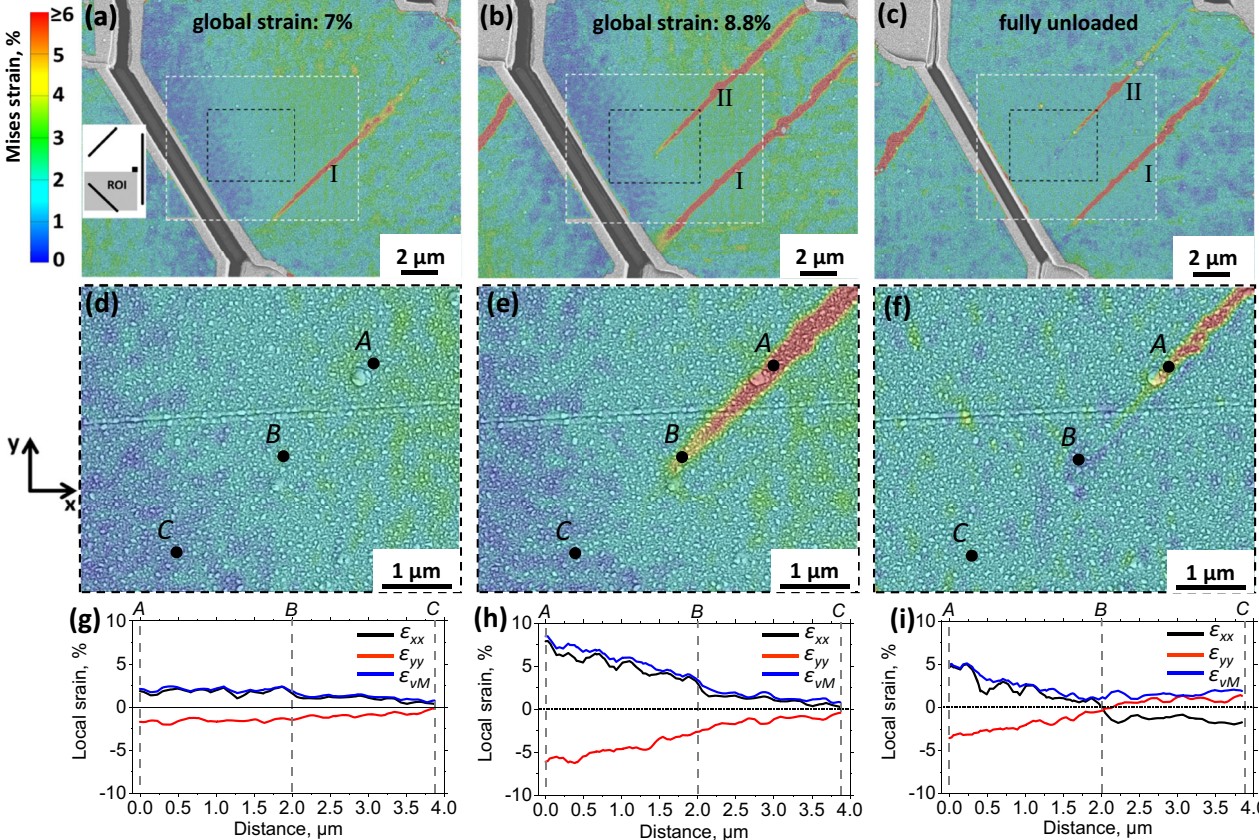

**Fig. 3 | Birth, arrest, and relaxation of SBs.** The distribution of von Mises strain within the N-A pattern after 7% global strain (**a**), after 8.8% global strain (**b**), and after full unload (**c**). The position of ROI is sketched in the inset of (**a**). The first and second generated SBs are marked by I and II, respectively. Enlarged views of the areas marked by black dashed rectangles in (**a**–**c**) are shown in (**d**–**f**), respectively. Black circles marked by $A$, $B$, and $C$ depict characteristic points within the SB, at its tip and ahead of the tip, respectively. The line profiles of the major, minor, and von Mises strains between points $A$ and $C$ for the corresponding three loading states are given in (**g**–**i**), respectively. The straining direction is horizontal.

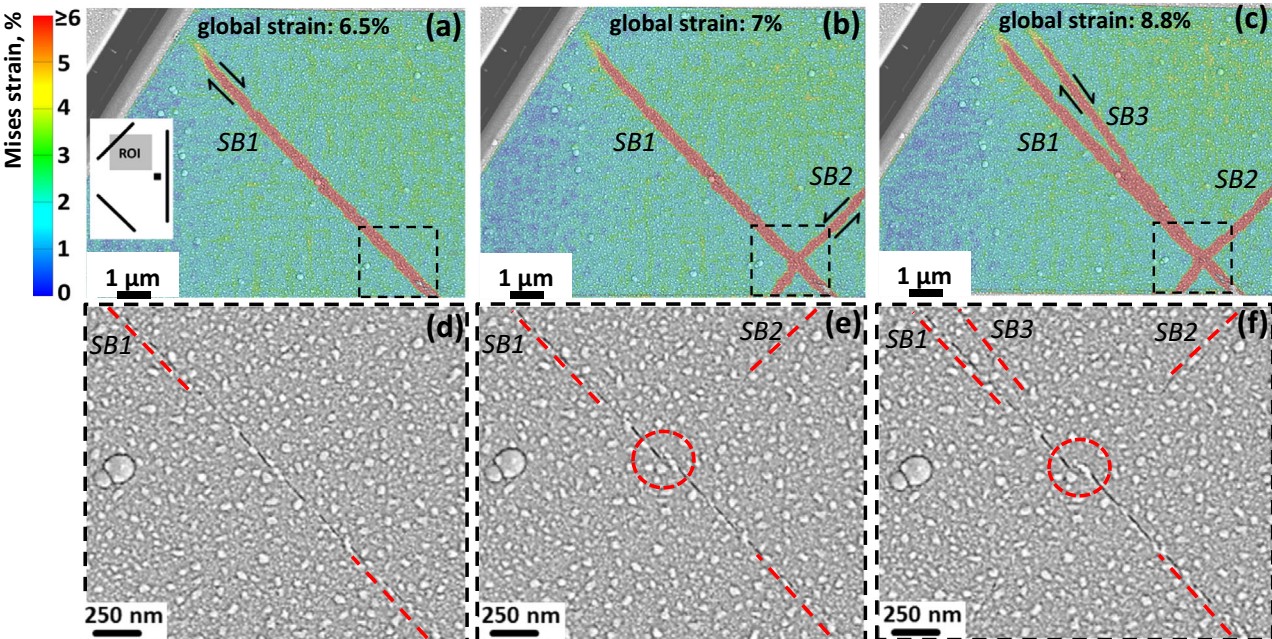

**Fig. 4 | Captured events of SB interaction and SB branching.** The von Mises strain maps are shown in (**a**–**c**) while the corresponding enlarged SEM images from the dashed rectangles are given in (**d**–**f**), respectively. At the global strain of 6.5% a single SB marked as "*SB1*" exists within the ROI (**a**, **d**). During the next step, a new SB marked as "*SB2*" crosses *SB1* at an angle of about 90° (**b**, **e**). During the next loading step, *SB3* appears from the point of intersection of *SB1* and *SB2* (**c**, **f**). The straining direction is horizontal. The inset in (**a**) depicts the approximate position of ROI. The red dashed circles in (**e**, **f**) mark the intersection point of *SB1* and *SB2*.

## Shear band interaction and splitting

Another characteristic event that is highly valuable for understanding the interactions between shear bands is depicted in Fig. 4. Within an N-A pattern SB (marked by "*SB1*") was initiated during the second loading step and has reached the state depicted in Fig. 4a after the applied global strain of 6.5%. During the next loading step another SB (marked by "*SB2*") crossed it at approximately 90° as shown in Fig. 4b. After the last loading step, the *SB1* split at the point of intersection with the *SB2* leading to the appearance of the new branch (marked by "*SB3*"), as depicted in Fig. 4c. The SEM images of the area around the intersection point of *SB1* and *SB2*, corresponding to the dashed rectangles in Fig. 4a–c, are shown in Fig. 4d–f, respectively. The *SB1* exhibits a distinct surface trace, additionally marked by the dashed red lines in Fig. 4d to guide the eye. The *SB2*, in contrast, is barely detectable on the SEM image without DIC analysis (Fig. 4e). The intersection point is marked by the red dashed circle in Fig. 4e. The disruption of the plane of *SB1* at the intersection point is more evident in Fig. 4f, at the same time, the *SB3* is also barely detectable with the used SEM imaging parameters. The video showing the evolution of all three independent components of the strain tensor is available as Supplementary video 3.

The appearance of the new *SB3* from the intersection point (Fig. 4c, f) suggests that a shear banding event can be understood as a kind of a wave which was generated at the stress concentrator, propagated within the initial plane of *SB1* and, at the intersection point, propagated further through the pristine material. This is a somewhat surprising observation which highlights the extreme localization of a shear banding phenomenon: disruption of *SB1* at the intersection point constitutes only a few tens of nanometers, still it was more favorable for the *SB3* to take a new path through the pristine material than bend back to the old path.

## Discussion

Having the information about local strain components at different stages of SB evolution, one can try to extract critical conditions required for the generation, propagation, and arrest of SBs. First, we would like to make the following two assumptions for the continuum mechanics model developed below: (i) the MG deforms purely elastically except within the areas of shear banding events and (ii) the plane stress condition (no force is acting in *z*-direction) adequately describes the system.

Since shear banding is the mechanism of plastic deformation and MGs are built on metallic bonds, the most natural choice of yielding criterion would be the von Mises criterion based on critical distortional energy. Although it was shown that macroscopic yielding of MGs can be adequately described by von Mises criterion[45–47], excellent agreement with the Mohr-Coulomb criterion was also reported[30,48,49]. The applicability of the Mohr-Coulomb criterion was, in turn, put into question, and the Rudnicki-Rice instability model was suggested instead[50]. For the considerations below, the exact form of the yield surface is not crucial, therefore, we employ the von Mises hypothesis stating that there exists a critical distortional energy density ($W_{ini}$) which can be carried by the material prior to plastic slip. We postulate that an SB is initiated within a small volume when local distortional energy density ($W_l$) reaches this critical value:

$$W_l = W_{ini} \qquad (1)$$

The condition (1) can be re-written in terms of strains giving:

$$\frac{1}{1+\nu}\sqrt{\frac{1}{2}\left[\left(\varepsilon_{xx}-\varepsilon_{yy}\right)^2+\left(\varepsilon_{yy}-\varepsilon_{zz}\right)^2+\left(\varepsilon_{xx}-\varepsilon_{zz}\right)^2\right]}=\varepsilon_{ini}, \qquad (2)$$

where $\nu$ is the Poisson's ratio, $\varepsilon_{xx}$, $\varepsilon_{yy}$, $\varepsilon_{zz}$ are the principal components of the strain tensor and $\varepsilon_{ini}$ is the critical equivalent strain for SB initiation. The critical strain $\varepsilon_{ini}$ cannot be measured directly within the current experimental approach, since DIC cannot resolve the strain in close proximity to the corner of a FIB cut where the SB is initiated. However, the critical strain can be estimated on the basis of the stress concentration factor (SCF). For a circular hole in infinite plate, the SCF equals 3[51] while for a square hole it is higher depending on the curvature of the corners[51]. To estimate the lower bound of $\varepsilon_{ini}$ we will use the

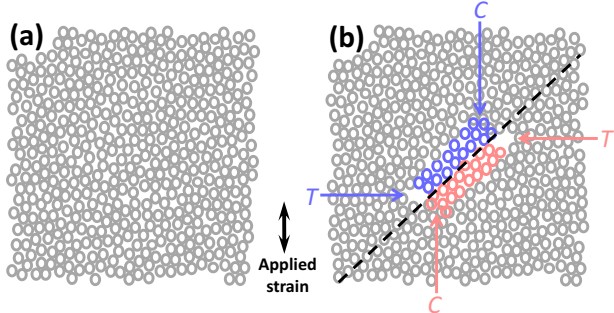

**Fig. 5 | Schematic description of SB initiation event on the atomistic scale.** The hypothetical atomic configurations immediately before and after an SB initiation event are shown in (**a**) and (**b**), respectively. The two groups of atoms taking part in shear slip in two opposite directions are marked by blue and red colors in (**b**). The slip leads to the appearance of local tension and compression areas, which are marked by the letters T and C, respectively. Atomic displacements within the *T* and *C* regions are exaggerated for better visibility. The black dashed line in (**b**) marks the shear plane along which the SB will propagate.

conservative value of SCF = 3 and assume linear elastic behavior, so that the same factor can be applied for strains. At SB initiation within the FIB patterns N (see Fig. 1b), the average strain within the areas symmetrically identical to the positions of the square cuts lies between 1.5% and 1.7% giving $4.5\% < \varepsilon_{ini} < 5.1\%$ (see Supplementary Fig. 1 for the details of the estimation procedure). This value is significantly higher than the macroscopic elastic limit of MGs but corresponds well to the estimations of the "ideal" elastic limit measured in micromechanical MG samples[52–54] or under the spherical indenter tip[55,56].

Using the same concept of critical distortional energy, one can formulate the condition of SB arrest:

$$W_l < W_{arr}, \tag{3}$$

stating that an SB is arrested if the local distortional energy density ahead of the SB tip is below the critical value $W_{arr}$. The corresponding values of $\varepsilon_{arr}$, i.e., the equivalent strain ahead of SB tip leading to SB arrest, measured to be in the range 1.5–1.9 % based on altogether ten experimentally observed SB arrest events. These values of local von Mises strain are surprisingly close to the macroscopic "universal" elastic limit of around 2% which is frequently reported for MGs.

The natural question arising now is what is the reason for such a difference between the critical strains for SB initiation and propagation? In order to tackle this issue, let us hypothesize about the atomistic mechanisms of SB initiation according to the Mises yielding criterion. Assume that a hypothetical atomic configuration depicted in Fig. 5a corresponds to the maximum distortional energy that can be carried by the material. If this maximum is exceeded in the middle of the volume (e.g., due to a local heterogeneity), then a sudden slip of two groups of atoms in opposite directions with respect to the slip plane occurs. These two displaced groups of atoms are marked by different colors in Fig. 5b. Evidently, to obey the first law of thermodynamics the slip must lead to relaxation, i.e., local distortion after slip is lower than before slip. Therefore, a mismatch between the relaxed atomic configuration within the slip area and elastically strained surroundings appears in the form of additional compressive and tension areas, according to Fig. 5b (the displacements of the atoms are enhanced for better visibility). Neighboring tension-compression areas can be also interpreted as additional shear imposed onto the slip plane. This additional shear superimposes with the apparent distortional strains and further kinetics depends on whether the total local strain exceeds the critical value $\varepsilon_{ini}$ (further SB propagation) or not (SB

arrest). Our model implies that, on the microscale, pristine MG always exhibits the "ideal" elastic limit. The limit is exceeded either due to stress concentration leading to SB initiation or due to the superposition of stored elastic strain with additional strain imposed by the propagating shear front. Shear band propagation through pristine material requires, therefore, fulfillment of two conditions simultaneously: the SB source must generate a shear banding event (i.e., Eq. (1) must be satisfied at the initiation site), and the local strain ahead of the SB tip must violate the arrest condition given by Eq. (3).

Developing this idea further, we claim that the initiation of a shear banding event at the source and spatial extension of the SB are two independent events. In other words, when a shear displacement is initiated at the source, it "doesn't know" how far it will propagate until the arrest condition at the tip of a propagating SB is probed. A clear manifestation of de-coupling between the initiation of a shear banding event and spatial extension of an SB was demonstrated in Fig. 4c, where a shear displacement initiated at the notch first propagated along the existing SB1 but then followed a new path through the pristine material until the arrest condition was satisfied within the stress relaxation zone in the vicinity of the FIB cut.

Real materials subjected to mechanical load always contain stress concentration areas caused by inclusions, casting defects (e.g., pores), surface features, or geometrical imperfections of the specimen. On the other hand, MGs were shown to be inhomogeneous also on the atomistic scale[57–59]. These atomistic heterogeneities have different local elastic properties[58] and, as a result, different critical elastic energy densities to initiate a SB. Within the current mesoscale approach, it is not possible to resolve the atomistic structure of the glass and directly correlate SB initiation events with local heterogeneities. At the same time, the presence of local heterogeneities does not contradict with the continuum mechanics approach. A heterogeneity per se cannot generate a shear band if the surrounding material is not distorted to a certain degree. Therefore, an increase in the local strain energy density is first required, and then local heterogeneities can be responsible for the exact atomistic position of the SB initiation event.

Provided experimental results and theoretical considerations allow us to propose the following mesoscopic model of shear banding in MGs. Applied mechanical load leads to an increase of elastic strain energy density within the material until the condition described by Eq. 1 is satisfied locally, at a stress concentration point. A pre-mature SB is then generated at a stress concentration point leading to propagation of plastic slip (Fig. 6a). If the distortional elastic strain energy density in the material ahead of a propagating SB is not high enough, then the SB will be arrested according to the condition given by Eq. 3 (Fig. 6b). Upon further loading, new shear banding events are induced from the stress concentration point and propagate along the SB plane leading to SB extension. Multiple slip events within the same SB bring it into a mature state, which is characterized by accumulation of plastic slip and resulting secondary effects of mechanical energy dissipation which could include dilatation, material extrusion, heating, voiding, cracking, etc. The dynamics of the mature SB operation is predominantly defined by the population of new weak spots, such as voids or nano- and microcracks, within the SB plain as well as by the reduction of traction force required to induce gliding of the two halves of the material due to broken atomic bonds and reduced material density (Fig. 6c). Since a mature SB could contain dozens or even hundreds of shear banding events, the adjacent material structure can be altered leading to formation of the shear affected zone. A mature SB can be therefore, considered as a 2D defect with lower activation energy for plastic slip in comparison to surrounding material, and it seems justified to consider the stage of mature SB operation separately from SB initiation and propagation through pristine material due to the different underlying mechanisms.

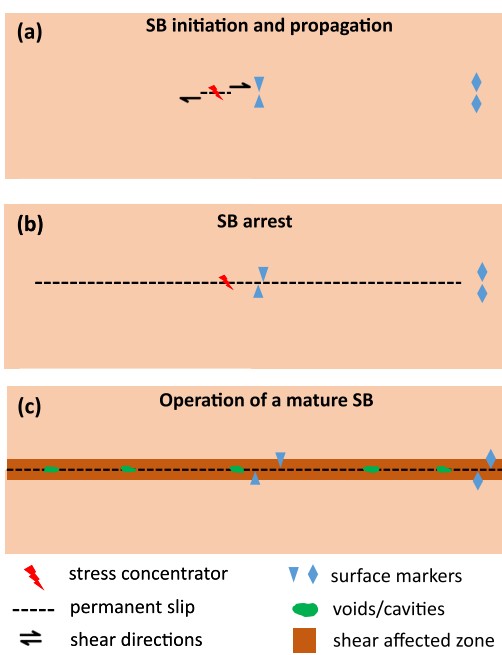

**Fig. 6 | Mesoscopic model of shear banding phenomenon. a** SB initiation stage where the condition given by Eq. 1 is satisfied locally at a stress concentrator. **b** propagation of a pre-mature SB through pristine material, (**c**) operation of a mature SB.

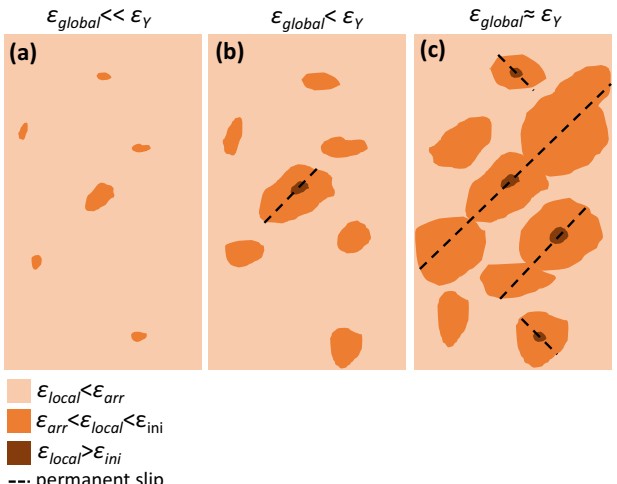

**Fig. 7 | Schematic explanation of the correlation between microscopic and macroscopic yielding of MGs.** Strain distribution within a MG specimen is inhomogeneous, and there are areas with local strain ($\varepsilon_{local}$) significantly higher than the applied global ($\varepsilon_{global}$) strain (**a**). First, SBs are initiated at strain concentration areas when the local strain exceeds the critical SB initiation strain ($\varepsilon_{ini}$), but they are arrested when encountering the areas with local strain below the critical strain ($\varepsilon_{arr}$) for SB propagation (**b**). Global yielding occurs when the first SBs can escape from the sample, inducing drops (serrations) in applied load (**c**).

A direct implication of the current model is that for initiation and propagation of a pre-mature SB no local structural changes prior to the shear banding event are required. The mechanisms of shear banding proposed in the literature are frequently based on the assumption that first the structure of the glass is changing locally (calling it collective STZ activation, rejuvenation, or cold dilatation), and then the actual permanent shear occurs[9,20,37,55,60,61]. It is important to remember that changes in local material structure within SBs reported in the literature

were detected after the corresponding shear banding events, typically using post-mortem high-resolution microscopy. We claim that local structural changes are secondary effects appearing as a consequence of rapid plastic shear deformation within an SB. In other words, plastic slip occurs first, and the dissipated energy can flow into local structural rejuvenation, void formation, material extrusion, acoustic emission, or heating.

In the frameworks of the suggested model, the seeming controversy between the global elastic limit of 2% and the significantly higher elastic limit observed in microscale experiments[52–54,62] can be easily explained according to the sketch shown in Fig. 7. Each macroscopic MG sample contains some defects (pores, cracks, inclusions, surface irregularities, etc.) which act as stress concentrators. Additional inhomogeneities of the strain distribution are caused by the mechanical contact between the sample and loading setup (gripping system, flat punch, indenter). As a result, some areas within the sample are under higher strain than the applied global strain (Fig. 7a). First SBs are initiated at stress concentration points with highest SCFs at global strains which are lower than the global elastic limit (Fig. 7b). Since the large fraction of sample volume is under strain which satisfies the arrest condition given by Eq. (3), these SBs are arrested while the load drops associated with the initiation events are too small to be captured by standard equipment and thus remain undetected. Global yielding (Fig. 7c) is observed when large enough volume fractions of the sample violate the arrest condition, and first, SBs can "escape" from the specimen, leading to a significant drop of applied load and formation of shear steps on the sample surface. Further loading leads to the activation of multiple SBs, which is reflected in multiple serrations in the stress-strain curves and the appearance of multiple steps on the sample surface.

Such a scenario implies that the universal elastic limit of MGs of about 2% does not describe the SB initiation condition or intrinsic material strength but corresponds to the violation of the SB arrest condition within a large enough volume fraction of the sample.

It is important to note that the macroscopic yielding models similar to the one presented in Fig. 7 were already suggested in the literature, although without the detailed characterization of local strains at the nanoscale. For instance, the progressive propagation of SBs captured in the interrupted compression experiments of macroscopic Pd-based and Ti-based MGs[14] was explained by successive initiation and arrest events. In another work, by using a load cell with a high acquisition rate (100 kHz), small serrations were shown to occur prior to large load drops in the compressive stress-strain curve[15]. The small serrations were attributed to the generation of pre-mature SBs within the sample before the major SB crossed the whole sample.

One of the long-standing controversy regarding the mechanisms of SB propagation is whether the slip occurs simultaneously[12,13,60,61,63], in a progressive manner[14,16,60,61,63], or these are two different modes of SB operation[15,60,61,63]. The model suggested in the present work can provide a consistent explanation of these seemingly controversial observations. Let us perform a thought experiment where an SB is generated at point A and propagates towards point B according to the sketch presented in Fig. 8. Let us also assume that shear displacement occurs purely within the figure plane and that there are surface traces (markers) in form of straight lines oriented perpendicularly to the SB plane (Fig. 8a). After being generated at A, the SB possesses the two major possibilities to reach B. If the condition for shear band arrest is violated across the whole distance between points A and B, then the SB will propagate at once, and all surface traces will exhibit the same shear displacement, as depicted in Fig. 8b. If, however, during loading the SB will be arrested and re-activated several times, then there will be a gradient in shear displacements of the surface traces, as depicted in Fig. 8c. We claim that when an SB was formed through a single shear banding event without intermediate arrest stages, then it was interpreted as "simultaneous slip" in previous reports. On the other hand,

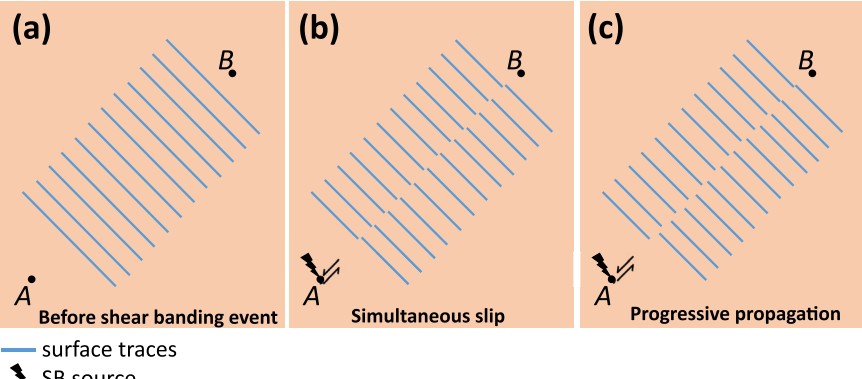

**Fig. 8 | A model for simultaneous and progressive propagation of SBs.** Depicted is a hypothetical MG specimen with marked points "A" and "B" as well as the surface traces between these points (**a**). When the load is applied, an SB is generated at A and extends towards B. If the SB arrest condition is violated across the whole distance AB, then the SB will propagate at once, and all surface traces will exhibit the same shear displacement, which can be interpreted as simultaneous slip (**b**). If the SB experiences several arrest and re-activation events during propagation from A to B, then the surface traces will exhibit a gradient in shear displacement: from maximal close to A to minimal close to B (**c**).

directly measured gradients of shear displacements along an SB[14,15] were induced by multiple arrest events during a mechanical test.

Therefore, our model implies that the SBs always propagate in a progressive manner, whereas well-documented different profiles of shear displacements along an SB simply reflect whether it was formed by a single slip or through multiple slip-arrest events.

The variety of structural changes within SBs which were reported in the literature (from almost no changes in HRTEM to voids and cracks)[24–28] can also be elucidated with the help of the suggested model. The atomistic structure within SB is typically characterized post-mortem using HRTEM. In most cases, well-detectable, mature SBs with unknown history and unknown local stress/strain conditions are considered. Moreover, global loading conditions to generate SBs for post-mortem HRTEM can be extremely different: cold rolling[25], high-pressure torsion[23], bending[27], compression[24,26,28]. We suppose that more mature SBs, after the higher amount of shear banding events, will inevitably exhibit more distinct structural changes up to voids and cracks, while pre-mature SBs would have nearly no structural difference to the bulk. Therefore, in further investigations, the information about the SB history should be extracted to guarantee a meaningful comparison of SB structure from experiments performed in different groups and under different conditions.

A promising research direction for the future design of MGs is a systematic analysis of the nature and distribution of SB sources in real materials. The collection of experimental data about the SCF values of typical stress concentrators, as well as the measurements of their density, should be the next step towards the advanced design of MG properties. Through proper adjustment of characteristics of SB sources, the macroscopic mechanical behavior can be tailored in a straightforward way. The lower the SCF associated with existing SB sources, the higher the measured macroscopic elastic limit will be. However, the material will behave in an extremely brittle manner since once an SB is initiated, it will immediately cross the sample. In contrast, if the SB sources possess a high SCF, then numerous SBs will be generated at low applied global strains being, however, unable to propagate thus contributing to extended ductility. We hypothesize that to gain both strength and ductility one would need to create dynamic SB sources which reduce their SCF after triggering a shear banding event. A MG with such dynamic sources should exhibit some ductility, potentially strong strain hardening effect, as well as high strength.

Our experiments confirm that no significant heating occurs during the early stage of SB formation. The melting point of bulk In is about 157 °C while the glass transition temperature of PdSi is about 370 °C[64,65]. Since no signs of any melting of the nanoislands in the vicinity of SBs were observed (e.g., compare Fig. 2g, f), the temperature within the SBs was far below the glass transition or crystallization temperature of PdSi. This is a further experimental confirmation that SB heating is a consequence of energy dissipation during large plastic slips and not a mechanism required for the initiation and propagation of SBs. It is important to note, that the measured absolute values of plastic slip in the considered system were of the order of a few tens of nanometers, which is orders of magnitude lower than the plastic slips reported in free-standing samples with a significant temperature rise[1–4]. Therefore, the absence of melting within the In nanoparticle coating is not surprising.

To sum up, we were able to capture the dynamic stages of SB propagation and map full-field strains within and around propagating SBs with a resolution of a few tens of nanometers. The conditions for SB initiation and SB arrest were formulated quantitatively in terms of critical local strain and von Mises yield criterion. One of the most fundamental implications of our work is that a pristine metallic glass always exhibits an elastic limit of about 5%, i.e., close to the theoretical limit. This elastic limit is either reached at a stress concentrator, which then acts as an SB initiation site, or through the superposition of the stored local strain tensor with the additional strain induced by the shear displacement wave on the tip of a propagating SB. It is directly demonstrated that a SB is arrested if the elastic strain ahead of the tip is below the certain value in the range 1.5−1.9% in terms of von Mises strain. Therefore, the universal elastic limit of about 2%, which is observed for a large variety of MGs corresponds to the violation of the SB arrest condition within large volumes of a specimen.

For further propagation of an arrested SB two conditions must be satisfied simultaneously: (i) the SB source must generate a shear banding event, and (ii) the local strain ahead of the SB tip must violate the arrest condition. The long-standing discussion about whether the slip occurs simultaneously or in a progressive manner can be easily resolved in the frameworks of the suggested model: the slip appears as simultaneous when the SB arrest condition is violated and the shear displacement can propagate over large distances or across the whole specimen. In this case, the shear displacement along the SB is virtually the same. If an SB during propagation experiences multiple arrest events, then there will be a gradient of shear displacements along the SB, which can be interpreted as progressive propagation.

The provided model of shear banding in metallic glasses is based purely on continuum mechanics and classical von Mises yielding criterion. Therefore, there is no need to assume the existence of a specific to metallic glasses structural state (called cold dilatation, rejuvenation, cooperative STZ activation) prior to slipping unless solid experimental

evidence that such a state really exists is provided. The continuum mechanics model suggests an alternative concept for the design of amorphous alloys: control over macroscopic strength and ductility should be achieved through control over the SB sources to tailor SB initiation events and control over the local strain fields to tailor SB propagation dynamics.

## Methods

$Pd_{71}Si_{29}$ thin films were deposited in a custom-built laboratory-scale unbalanced dc magnetron sputtering system by co-sputtering from two elementary targets (Pd, 99.95% purity; Si, 99.999% purity) with a base pressure $< 10^{-4}$ Pa. Deposition was performed in Ar atmosphere with pressure of 0.4 Pa without substrate heating. The target power was set to 60 W at the Pd target and 140 W at the Si target, resulting in a deposition rate of 0.7 nm/s. The substrate material, a polyimide Upilex® foil ($50 \times 50$ mm²) with a thickness of 50 μm and single-crystalline (001) Si substrates with a thickness of 350 μm for thickness measurements, were mounted on a rotatable sample holder opposite to the targets with a target-to-substrate distance of 40 mm. The substrates were cleaned in an ultrasonic bath of ethanol and blown dry with dry air immediately before loading them into the vacuum system. Prior to deposition, the substrates were plasma-etched for 2 min to clean the surface and promote adhesion. Thin film deposition was done on grounded substrates, without applying external heating.

Since the sputter deposition process is physically different from the rapid quenching of the melt, the energetic state of the obtained glass cannot be easily defined. It is particularly important to avoid a highly rejuvenated state which may exhibit significantly higher plasticity due to increased free volume. A highly rejuvenated state of the considered PdSi films can be excluded due to the following reasons. (i) Mechanical tests described in this work were performed about 10 months after the deposition. Even if there was a higher energetic state after the deposition, the room temperature relaxation process must have occurred during this time. (ii) Since the polymer substrate is much more compliant than the glass, even small residual stresses within the film immediately manifest themselves by a substrate curvature. The curvature of the substrate was never observed for the PdSi system, neither directly after the deposition nor after months of shelf time, indicating that no significant changes in the free volume have occurred.

The chemical composition of the films, determined by energy-dispersive X-ray spectroscopy (EDX) in a Zeiss Leo 1525 SEM, was within the range $71 \pm 2$ at% Pd and $29 \pm 2$ at% Si. The film thickness of $400 \pm 10$ nm was determined by examining the cleaved cross-section of the film deposited on the Si substrate in SEM.

The X-ray diffraction analysis, HRTEM characterization, and density measurements by X-ray reflectivity of similar sputter-deposited PdSi films were published elsewhere, and the reader is referred to these publications for more details[42,66].

The speckle pattern was created by sputter deposition of pure In on the surface of PdSi film employing the In target with a purity of 99.99%, deposition time of 20 s, applied power of 20 W, and Ar pressure of 0.8 Pa.

Sample pre-patterning was conducted by FIB-milling on a Zeiss Auriga dual beam workstation equipped with a Ga ion gun. Seven microscopic patterns were milled within a single macroscopic sample of dimensions $4 \times 40$ mm². The macroscopic tensile strain was applied by means of a custom-made compact manual straining device. The applied global strain was measured directly on the sample surface by measuring the distance between two markers at low magnification (300x) at each loading step. In this paper, we consider a mechanical loading profile consisting of seven global loading steps ($0\% - 2\% - 4.7\% - 5.6\% - 6.5\% - 7\% - 7.5\% - 8.8\%$ of total global strain) and two unloading steps (5.7 % – fully unloaded) applied in-situ in a scanning electron microscope (SEM).

For DIC analysis, at each loading step SEM images of corresponding ROIs were taken from each pre-patterned area, therefore, the local strain distributions revealed by DIC for different patterns correspond to the same applied profile of global strain. As a result, the specimen stayed for a time of about 1 h at each loading step in order to acquire multiple images. It is also important to mention that during this holding time the sample was kept at constant displacement (strain). It could be important when comparing the current approach to load-controlled mechanical tests of free-standing specimens. The SEM images were taken at magnifications of 12 kx and 24 kx with a resolution of $1024 \times 768$ pixels, leading to the pixel size of 26 nm and 13 nm, respectively.

DIC was performed using the free version of GOM Correlate 2016 software. For the images taken at the magnification of 24 kx (Figs. 2–4) the subset size of 15 pixels and step size of 5 pixels were used. For the images taken at the magnification of 12 kx (Fig. 3), the subset and step sizes were 25 pixels and 6 pixels, respectively. Although DIC, in general, is able to detect sub-pixel displacements[67], when estimating the spatial resolution of the strain maps considered in the current work, we stick to a conservative value which is equal to the step size, i.e., 65 nm for the images taken at a magnification of 24 kx, and 156 nm for the images taken at the magnification of 12 kx (Fig. 3a–c).

## Data availability

Raw SEM images corresponding to Fig. 2a–d are provided in supplementary data. Further data sets generated during the current study are available from the corresponding author on request. Source data are provided with this paper.

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

## Acknowledgements

O.G. would like to acknowledge full financial support from the Austrian Science Fund (FWF) and Land Steiermark, Project No. P31544-NBL.

## Author contributions

**Oleksandr Glushko**: Conceptualization, Methodology, Validation, Formal analysis, Investigation, Writing - Original Draft, Writing - Review & Editing, Visualization, Funding acquisition; **Reinhard Pippan**: Conceptualization, Methodology, Visualization, Writing - Review & Editing; **Daniel Sopu**: Conceptualization, Writing - Review & Editing; **Christian Mitterer**: Writing - Review & Editing; **Jürgen Eckert**: Writing - Review & Editing;

## Competing interests

The authors declare no competing interests.
