## [Peer Review File · Nature Communications]

How to catch a shear band and explain plasticity of metallic glasses with continuum mechanicsREVIEWER COMMENTS

Reviewer #1 (Remarks to the Author):

Comments on the text entitled

"Nanoscale full-field strains around propagating shear bands and multiscale continuum mechanics model of plasticity in metallic glasses"

By O. GLUSHKO et al

Submitted to Nature Com.

The question of the deformation of metallic glasses at room temperature is a topical issue and the subject of numerous publications. The text presented here is set in this context. Using very fine measurements of electron microscopy images, the authors were able to highlight the mechanisms of initiation and propagation of shear bands in a thin film of Pd-Si binary metallic glass. A continuum mechanics model was used to interpret their results.

The results are extremely convincing.

The text is clear and well written.

The English language seems to me to be quite correct.

The references are numerous and cover the field well.

However, I have a few questions or comments:

1- The results are obtained on a particular binary alloy. As the authors mention, the composition of the alloy can have an influence on the deformation mechanism itself. For some alloys, there seems to be an increase in temperature at the shear bands. For others, no. What do the authors think?

2- Recent studies show that amorphous materials are not homogeneous but have significant local heterogeneities (e.g. J. Qiao et al, Prog. Mater. Sci (2019)). These heterogeneities in chemical composition, elastic characteristics.... certainly have an influence on the initiation of shear bands. This point needs to be discussed.

3- Furthermore, does the existence of these heterogeneities allow a correct application of continuum mechanics?

4- It seems to me that experiments involving partial, or even progressive, stress recovery could provide interesting information on the return mechanisms.

In conclusion, this is an extremely interesting study, but it needs to be modified before it can be published.

Reviewer #2 (Remarks to the Author):

This paper is excellent. It presents an original method to analyze the dynamics of shear bands in metallic glasses achieving an amazing insight into the mechanisms of initiation and propagation of shear bands. I think the work is highly original and relevant. In my opinion it may have an important impact on our understanding of the plasticity mechanisms in metallic glasses. It deserves publication in Nature Communications.

However, I suggest the authors to consider the following points for the resubmitted version:

In what position within the Potential Energy Landscape (PEL) the sputter-deposited metallic glass is situated. It should be considered a high energy configuration or rather a relatively stable one? This is important to know, as the mechanisms of shear band initiation and propagation so well visualized in the paper may be not general and, on the contrary, be particular of a certain degree of structural relaxation. Density, X-ray diffraction or calorimetric measurements could be used to determine if the glass configuration can be considered as rejuvenated or relaxed. In my opinion, the important point is that the authors should demonstrate that the glass configuration can be considered as 'normal', i.e. the thin metallic glass film it is not in and extremely rejuvenated or relaxed structure, which could then behave very differently from the usual materials produced by casting methods.

I cannot find in the methods how long the samples stay in each strain step. I think the authors should provide this information. Although at room temperature it is expected that the anelastic and homogenous plastic relaxation will be small, glasses are expected to have large contribution of anelastic relaxation if the time is long enough. Also, we do not know the position of the sputtered structure in the PEL, and for high energy structures the effects of anelasticity (time delayed elasticity) could be important even at room temperature. I think this information may be useful for other groups trying to reproduce similar experiments.

I do not completely understand fig. 5. What is the direction of propagation of the initiated SB? Should not be another group of atoms shearing in opposite direction? In my opinion the authors should explain better what they try to represent in fig. 5.

Section 4.7, concerning the discussion of the possible temperature rise in the monitored shear bands. In this point I think the authors should try to estimate what amount of heat would be necessary to melt the Indium islands and what amount of heat would be necessary to melt the material within the shear band. It seems to me a rather complex estimation which involves size of Indium islands, the quantity of sheared material, the duration of the shear band propagation, the heat flow due to conduction, etc. I am not sure that a locally increase of temperature in the shear band can be so easily discarded. I agree with the authors that their results point to such conclusion but I think they cannot absolutely demonstrate it.

Some small typos and errors should be corrected:

Line 244: 'then bend back' -> 'than bend back'

Line 371: 'within the large' -> 'within a large'

Manuscript Number: **NCOMMS-23-54992**

Title: "Nanoscale full-field strains around propagating shear bands and multiscale continuum mechanics model of plasticity in metallic glasses"

Authors' reply to reviewers' comments.

Reviewer#1

The authors would like to thank the reviewers for careful reading the manuscript as well as for investing time in detailed feedback about the shortcomings and potential improvements. It is of course very pleasant to hear that performed research and the resulting manuscript are found interesting and relevant.

A detailed reply to each expressed concern is given below along with the changes/additions to the manuscript itself, whereby:

- *italic* font marks original reviewers questions;
- standard black font corresponds to authors' reply;
- **dark green** font depicts the parts of the manuscripts which were added or changed (both here and in the manuscript file);

"1- The results are obtained on a particular binary alloy. As the authors mention, the composition of the alloy can have an influence on the deformation mechanism itself. For some alloys, there seems to be an increase in temperature at the shear bands. For others, no. What do the authors think?"

The question of the temperature rise within the shear bands is one of the very controversial issues in metallic glass community. In the famous Nature Materials paper by Lewandowski and Greer (Lewandowski, J. J. & Greer, A. L. Temperature rise at shear bands in metallic glasses. *Nat. Mater.* **5**, 15–18 (2006).) the temperature rise within the shear bands of up to few thousand kelvin was claimed based on the observation of melted tin coating in the vicinity of surface cracks. However, several research groups (see e.g. Slaughter, S. K. *et al.* Shear bands in metallic glasses are not necessarily hot. *APL Mater.* **2**, (2014) or Refs. 6-11 in the manuscript) have convincingly shown that temperature rise is observed after sample fracture but is not observed if the loading stopped before fracture despite numerous generated shear bands. It is important to mention that the absence of significant temperature rise was shown for the similar CuZr-based metallic glasses as in the original paper by Lewandowski and Greer but also for other alloys with different glass transition temperatures. In our opinion, it seems to be established that temperature rise can be observed as a result of energy dissipation during fracture or significant sudden slip but it is not required for initiation and propagation of shear bands per se. Absence of any melting of indium coating (which has even lower melting temperature than tin) on top of a shear band in our manuscript confirms this statement. It is necessary to note, however, that we have captured rather early stages of SB evolution with total displacements of a 100-200 nanometers occurred in dozens of single slip events. Due to the polymer support, large sudden slips, as in free-standing samples, would not be possible in our experimental system, therefore absence of heating effects is not surprising.

To further address this issue, we have performed some calculations according to [Lewandowski, J. J. & Greer, A. L. Temperature rise at shear bands in metallic glasses. *Nat. Mater.* **5**, 15–18 (2006)]. It is important to mention that there is a very important unknown parameter in this kind of temperature

estimations - the actual amount of generated heat. In Lewandowski&Greer paper the heat content within a SB was estimated based on the width of molten tin coating and temperature rise required to melt tin. Since there is no molten indium in current experiments, we can only estimate the upper bound of the generated heat and state that it was less in our PdSi system.

For the estimation of the temperature rise within the shear band, Eq. (4) from the Lewandowski&Greer paper was utilized:

$$\Delta T_{\text{centre}} = \frac{1}{\sqrt{\pi}} \left(\frac{H}{\rho C} \right) \sqrt{\frac{1}{\alpha \delta t}} = \frac{1}{\sqrt{\pi}} \left(\frac{H}{\rho C} \right) \sqrt{\frac{V}{\alpha y}}, \quad (4)$$

where ΔT_{centre} is the temperature inside the shear band immediately after the slip, H is the heat content, ρ is the material density, C is specific heat capacity, V is the velocity of the slip, α is the thermal diffusivity and y is the shear offset. The following material parameters were used in the calculations:

$C=368 \text{ J}/(\text{kg} \cdot \text{K})$ (<https://doi.org/10.1088/0256-307X/29/4/046402>)

$\alpha = 2 \times 10^{-6} \text{ m}^2/\text{s}$ ([https://doi.org/10.1016/S1359-6462\(02\)00160-4](https://doi.org/10.1016/S1359-6462(02)00160-4))

$\rho = 10160 \text{ kg}/\text{m}^3$ (own XRR measurement)

$y = 30 \text{ nm}$ (current manuscript)

The slip velocity is the second big unknown. The maximum slip velocity equals to the speed of transverse sound wave (1700 m/s, <https://10.1016/j.jallcom.2010.09.065>). However, the actual slip velocity could be much lower, therefore, for the second limiting case of estimation the value of 1% of the sound velocity was taken (17 m/s).

In order to reach the temperature increase within the SB of 200°C (to melt the tin coating for sure), the heat content of about 80 J/m² is needed assuming the slip velocity of 17 m/s and about 8 J/m² if the slip occurs with the speed of sound. These values are, expectedly, significantly lower than those estimated by Lewandowski&Greer (400 – 2200 J/m²).

To authors opinion, these estimations would not make any significant contribution to the paper. The experimental fact that no melting of indium coating above the SB is observed implies that the amount of generated heat is low. This, in turn, is not surprising owing to the directly measured shear displacements of just a few tens of nanometers (in contrast to about 2 μm in Lewandowski&Greer paper). Therefore, direct comparison to the case of molten tin in Lewandowski&Greer paper, or to other cases where free-standing samples with large slips were examined, seems to be pointless. Moreover, the question of correlations between the amount of slip and temperature rise was already addressed in numerous papers (see Refs. [5-11] of the manuscript) using experimental designs and equipment which are much better suitable for this task. Therefore, the following text is added to the manuscript (Section 4.7) as a summary of this discussion:

It is important to note, that the measured absolute values of plastic slip in the considered system were of the order of a few tens of nanometers which is orders of magnitude lower than the plastic

slips reported in free-standing samples with a significant temperature rise¹⁻⁴. Therefore, the absence of melting within the In nanoparticle coating is not surprising.

“2- Recent studies show that amorphous materials are not homogeneous but have significant local heterogeneities (e.g. J. Qiao et al, Prog. Mater. Sci (2019). These heterogeneities in chemical composition, elastic characteristics.... certainly have an influence on the initiation of shear bands. This point needs to be discussed.

3- Furthermore, does the existence of these heterogeneities allow a correct application of continuum mechanics?”

It is a very good point. Indeed, the concept of structural heterogeneity of metallic glasses is widely known and is well-accepted in the community due to numerous experimental and computational evidences which are provided in the literature. Within our mesoscale SEM-based approach we are not able to look into the atomistic structure and therefore cannot directly correlate local heterogeneities with SB initiation events. At the same time, the presence of local heterogeneities does not contradict with the continuum mechanics approach. A heterogeneity per se cannot generate a shear band if surrounding material is not distorted to a certain degree. Local heterogeneities, either of chemical or structural nature, must possess also local variation of elastic properties and, as a result, local variation of critical strain energy density to initiate a SB. Therefore, these heterogeneities might indeed be responsible for specific positioning of the SB initiation event on sub-nm scale, but first a certain value of elastic (distortional) strain energy density must be stored in the material. To discuss this question raised by the Reviewer, the following text was added to the manuscript (along with three additional references):

Real materials subjected to mechanical load always contain stress concentration areas caused by inclusions, casting defects (e.g. pores), surface features or geometrical imperfections of the specimen. On the other hand, MGs were shown to be inhomogeneous also on the atomistic scale⁵⁷⁻⁵⁹. These atomistic heterogeneities have different local elastic properties⁵⁸ and, as a result, different critical elastic energy density to initiate a SB. Within the current mesoscale approach, it is not possible to resolve the atomistic structure of the glass and directly correlate SB initiation events with local heterogeneities. At the same time, the presence of local heterogeneities does not contradict with the continuum mechanics approach. A heterogeneity per se cannot generate a shear band if the surrounding material is not distorted to a certain degree. Therefore, an increase in the local strain energy density is first required and then local heterogeneities can be responsible for the exact atomistic position of the SB initiation event.

“4- It seems to me that experiments involving partial, or even progressive, stress recovery could provide interesting information on the return mechanisms.”

Indeed ultra-slow strain rate or recovery experiments are really required to understand the flow capabilities of glasses, however, in our case the substrate would greatly influence the recovery dynamics due to the viscoelasticity of the polymer. The usage of free-standing nanoscale samples combined with in-situ TEM would be required, and research work in this directions is definitely being performed, see, e.g. [Idrissi, H., Ghidelli, M., Béché, A. et al. Atomic-scale viscoplasticity mechanisms

revealed in high ductility metallic glass films. Sci Rep 9, 13426 (2019).
<https://doi.org/10.1038/s41598-019-49910-7>].

Manuscript Number: **NCOMMS-23-54992**

Title: "Nanoscale full-field strains around propagating shear bands and multiscale continuum mechanics model of plasticity in metallic glasses"

Authors' reply to reviewers' comments.

Reviewer#2

The authors would like to thank the reviewers for careful reading the manuscript as well as for investing time in detailed feedback about the shortcomings and potential improvements. It is of course very pleasant to hear that performed research and the resulting manuscript are found interesting and relevant.

A detailed reply to each expressed concern is given below along with the changes/additions to the manuscript itself, whereby:

- *italic* font marks original reviewers questions;
- standard black font corresponds to authors' reply;
- **dark green** font depicts the parts of the manuscripts which were added or changed (both here and in the manuscript file);

"In what position within the Potential Energy Landscape (PEL) the sputter-deposited metallic glass is situated. It should be considered a high energy configuration or rather a relatively stable one? This is important to know, as the mechanisms of shear band initiation and propagation so well visualized in the paper may be not general and, on the contrary, be particular of a certain degree of structural relaxation. Density, X-ray diffraction or calorimetric measurements could be used to determine if the glass configuration can be considered as rejuvenated or relaxed. In my opinion, the important point is that the authors should demonstrate that the glass configuration can be considered as 'normal', i.e. the thin metallic glass film it is not in and extremely rejuvenated or relaxed structure, which could then behave very differently from the usual materials produced by casting methods."

This is a good and complicated question. We believe that our PdSi glass is rather in a normal state and for sure not highly rejuvenated. First of all, significant time has passed between the synthesis of the PdSi films and mechanical tests. Specifically, for the sample batch considered in the manuscript, 10 months has passed after the deposition (approximately 2.6×10^7 s). During this time a potential rejuvenated state must relax. This same sample batch was used several time during preparation and methodology optimization before, but also after this time and we never observed any changes in the material behavior. Secondly, the same deposition recipe was applied dozens of times to fabricate PdSi films with different thicknesses (between 7 nm and 1500 nm) and large polymer substrates 50x50 mm² were utilized. Even for the thickest films no substrate curvature after deposition was ever observed. Also with time we never observed visible curvature of the substrates. It means that (i) there are virtually no residual stresses after deposition and (ii) no detectable volume change (as a result of density increase due to relaxation) has occurred during the observation time.

We indeed have the measurement of material density (from X-ray reflectivity measurements of 100 nm thick PdSi film), we have flash-DSC measurements and also X-ray diffractograms. However, we never worked on relaxation/rejuvenation treatments of this material and therefore do not have two distinct states in terms of PEL position to compare them.

The following discussion is added to Materials and Methods section:

Since the sputter deposition process is physically different from the rapid quenching of the melt, the energetic state of the obtained glass cannot be easily defined. It is particularly important to avoid a highly rejuvenated state which may exhibit significantly higher plasticity due to increased free volume. A highly rejuvenated state of the considered PdSi films can be excluded due to the following reasons. (i) Mechanical tests described in this work were performed about 10 months after the deposition. Even if there was a higher energetic state after the deposition, room temperature relaxation process must have had occurred during this time. (ii) Since the polymer substrate is much more compliant than the glass, even small residual stresses within the film immediately manifest themselves by a substrate curvature. Curvature of the substrate was never observed for the PdSi system, neither directly after the deposition nor after months of shelf time, indicating that no significant changes in the free volume have occurred.

"I cannot find in the methods how long the samples stay in each strain step. I think the authors should provide this information. Although at room temperature it is expected that the anelastic and homogenous plastic relaxation will be small, glasses are expected to have large contribution of anelastic relaxation if the time is long enough. Also, we do not know the position of the sputtered structure in the PEL, and for high energy structures the effects of anelasticity (time delayed elasticity) could be important even at room temperature. I think this information may be useful for other groups trying to reproduce similar experiments."

It is indeed important information which is missing. The sample was kept for about one hour at each step in order to acquire dozens of SEM images for DIC analysis (there were 24 microscopic ROIs in one macroscopic sample). The following text is added/modified in Materials and methods section:

For DIC analysis, at each loading step SEM images of corresponding ROIs were taken from each pre-patterned area, therefore, the local strain distributions revealed by DIC for different patterns correspond to the same applied profile of global strain. As a result, the specimen stayed for a time of about 1 hour at each loading step in order to acquire multiple images. It is also important to mention that during this holding time the sample was kept at constant displacement (strain). It could be important when comparing current approach to load-controlled mechanical tests of free-standing specimens.

The following statement was added to the Experimental approach section:

The sample was kept for a time of more than one hour at each step in order to acquire numerous SEM images.

"I do not completely understand fig. 5. What is the direction of propagation of the initiated SB? Should not be another group of atoms shearing in opposite direction? In my opinion the authors should explain better what they try to represent in fig. 5."

The authors are very grateful for this comment. Indeed, the Fig. 5 was not thought through. The intention was to depict the hypothetical atomistic condition on the tip of a propagating SB demonstrating the imposition of additional distortion on the material. Fig. 5 is now completely re-designed and shows that two groups of atoms are shearing in opposite direction thus imposing

additional shear on the slip plane. The discussion of Fig. 5 as well as the figure caption were re-written accordingly:

The natural question arising now is what is the reason for such a difference between the critical strains for SB initiation and propagation? In order to tackle this issue, let us hypothesize about the atomistic mechanisms of SB initiation according to the Mises yielding criterion. Assume that a hypothetical atomic configuration depicted in Fig. 5a corresponds to the maximum distortional energy which can be carried by the material. If this maximum is exceeded in the middle of the volume (e.g. due to a local heterogeneity) then sudden slip of two group of atoms in opposite directions with respect to the slip plane occurs. These two displaced groups of atoms are marked by different colors in Fig. 5b. Evidently, to obey the first law of thermodynamics the slip must lead to relaxation, i.e. local distortion after slip is lower than before slip. Therefore, a mismatch between the relaxed atomic configuration within the slip area and elastically strained surroundings appears in form of additional compressive and tension areas, according to Fig. 5b (the displacements of the atoms are enhanced for better visibility). Neighboring tension-compression areas can be also interpreted as additional shear imposed onto the slip plane. This additional shear superimposes with the apparent distortional strains and further kinetics depends on whether the total local strain exceeds the critical value ϵ_{ini} (further SB propagation) or not (SB arrest).

“Section 4.7, concerning the discussion of the possible temperature rise in the monitored shear bands. In this point I think the authors should try to estimate what amount of heat would be necessary to melt the Indium islands and what amount of heat would be necessary to melt the material within the shear band. It seems to me a rather complex estimation which involves size of Indium islands, the quantity of sheared material, the duration of the shear band propagation, the heat flow due to conduction, etc. I am not sure that a locally increase of temperature in the shear band can be so easily discarded. I agree with the authors that their results point to such conclusion but I think they cannot absolutely demonstrate it.”

Indeed, as mentioned in the Introduction, temperature increase is a long-standing discussion point in the metallic glass community. From our point of view, the main obstacle in every estimation of the temperature rise is the unknown parameters and large spread of possible or guessed values of these parameters. For instance, the estimations of shear band speed differ by orders of magnitude leading to temperature estimation which also differ by orders of magnitude (e.g. see Refs [5,9] of the manuscript).

In order to address the question of temperature rise in more details, we have performed some calculations according to [Lewandowski, J. J. & Greer, A. L. Temperature rise at shear bands in metallic glasses. *Nat. Mater.* **5**, 15–18 (2006)]. It is important to mention, that there is another very important unknown parameter in this kind of temperature estimations - the actual amount of generated heat. In Lewandowski&Greer paper the heat content within a SB was estimated based on the width of molten tin coating and temperature rise required to melt tin. Since there is no molten indium in current experiments, we can only estimate the upper bound of the generated heat and state that it was less in our PdSi system.

For the estimation of the temperature rise within the shear band, Eq. (4) from the Lewandowski&Greer paper was utilized:

$$\Delta T_{\text{centre}} = \frac{1}{\sqrt{\pi}} \left(\frac{H}{\rho C} \right) \sqrt{\frac{1}{\alpha \delta t}} = \frac{1}{\sqrt{\pi}} \left(\frac{H}{\rho C} \right) \sqrt{\frac{V}{\alpha y}}, \quad (4)$$

where ΔT_{centre} is the temperature inside the shear band immediately after the slip, H is the heat content, ρ is the material density, C is specific heat capacity, V is the velocity of the slip, α is the thermal diffusivity and y is the shear offset. The following material parameters were used in the calculations:

$C=368 \text{ J/(kg}\cdot\text{K)}$ (<https://doi.org/10.1088/0256-307X/29/4/046402>)

$\alpha = 2 \times 10^{-6} \text{ m}^2/\text{s}$ ([https://doi.org/10.1016/S1359-6462\(02\)00160-4](https://doi.org/10.1016/S1359-6462(02)00160-4))

$\rho=10160 \text{ kg/m}^3$ (own XRR measurement)

$y=30 \text{ nm}$ (current manuscript)

The slip velocity is the second big unknown. The maximum slip velocity equals to the speed of transverse sound wave (1700 m/s, <https://10.1016/j.jallcom.2010.09.065>). However, the actual slip velocity could be much lower, therefore, for the second limiting case of estimation the value of 1% of the sound velocity was taken (17 m/s).

In order to reach the temperature increase within the SB of 200°C (to melt the tin coating for sure), the heat content of about 80 J/m² is needed assuming the slip velocity of 17 m/s and about 8 J/m² if the slip occurs with the speed of sound. These values are, expectedly, significantly lower than those estimated by Lewandowski&Greer (400 – 2200 J/m²).

To authors opinion, these estimations would not make any significant contribution to the paper. The experimental fact that no melting of indium coating above the SB is observed implies that the amount of generated heat is low. This, in turn, is not surprising owing to the directly measured shear displacements of just a few tens of nanometers (in contrast to about 2 μm in Lewandowski&Greer paper). Therefore, direct comparison to the case of molten tin in Lewandowski&Greer paper, or to other cases where free-standing samples with large slips were examined, seems to be pointless. Moreover, the question of correlations between the amount of slip and temperature rise was already addressed in numerous papers (see Refs. [5-11] of the manuscript) using experimental designs and equipment which are much better suitable for this task. Therefore, the following text is added to the manuscript (Section 4.7) as a summary of this discussion:

It is important to note, that the measured absolute values of plastic slip in the considered system were of the order of a few tens of nanometers which is orders of magnitude lower than the plastic slips reported in free-standing samples with a significant temperature rise [1–4]. Therefore, the absence of melting within the In nanoparticle coating is not surprising.

REVIEWERS' COMMENTS

Reviewer #1 (Remarks to the Author):

Comments on the revised version of the text entitled
"Nanoscale full-field strains around propagating shear bands and multiscale continuum mechanics model of plasticity in metallic glasses"

By O. GLUSHKO et al

Submitted to Nature Com.

The question of the deformation of metallic glasses at room temperature is a topical issue and the subject of numerous publications. The text presented here is set in this context. Using very fine measurements of electron microscopy images, the authors were able to highlight the mechanisms of initiation and propagation of shear bands in a thin film of Pd-Si binary metallic glass. A continuum mechanics model was used to interpret their results.

The results are extremely convincing.

The text is clear and well written.

The English language seems to me to be quite correct.

The references are numerous and cover the field well.

The authors answered in a very satisfactory way to the questions, comments or criticisms included in my previous report.

In conclusion, this is an extremely interesting study, and I recommend TO ACCEPT the revised version for publication.

Reviewer #2 (Remarks to the Author):

Dear editor,

As already recommended in the previous round I think the paper deserves publication. I have just one minor point for the authors:

In the response letter, the authors say that they have Flash-DSC measurements, X-ray diffraction and density measurements of the sputtered metallic glass films. I think the authors should provide these results in supplementary material.